# Description of Process and Content of Online Dementia Coaching for Family Caregivers of Persons with Dementia

**DOI:** 10.3390/healthcare7010013

**Published:** 2019-01-19

**Authors:** Rita A. Jablonski, Vicki Winstead, David S. Geldmacher

**Affiliations:** 1School of Nursing, University of Alabama at Birmingham, Birmingham, AL 35294, USA; vickiwin@uab.edu; 2School of Medicine, University of Alabama at Birmingham; Birmingham, AL 35294, USA; dgeldmacher@uabmc.edu

**Keywords:** dementia, caregiver, coaching, rejection/resistance to care behavior, internet-based

## Abstract

Family caregivers of persons with dementia encounter resistance to care behaviors (RCBs). The purpose of this methods paper was to describe the process and content of six weekly 60-min caregiver coaching sessions delivered synchronously through an online platform to 26 family caregivers of persons with dementia. All session notes were analyzed for process; two coaching sessions from five purposely-selected participants were transcribed and analyzed thematically for content. The six sessions followed an overall pattern. The first session included the most teaching and goal-setting; the coaches also queried the family caregiver about the premorbid personality, work history, and interpersonal attributes of the person with dementia. Sessions two through five were the most active coaching sessions; previously suggested strategies were evaluated and tailored; caregivers also role-played with the coaches and developed scripts designed to curtail RCB. The sixth session served as a review of successful caregiver strategies and concluded the coaching relationship. Four primary content themes emerged in the coaching process: (1) education; (2) caregiver communication; (3) affirmation of the caregiver; and (4) individualized strategies. These four content categories were used throughout the coaching process and were interwoven with each other so that the participant knew why the behavior was occurring, how to verbally address it, how to use a strategy effectively, and affirmation of the result. The coaching process and content demonstrated alignment with person-centered practices and relationship-centered care.

## 1. Introduction

In the United States, 16 million informal caregivers provide 18 billion hours of care to 5.5 million persons with Alzheimer’s disease or related dementias annually; this care is valued at $230 billion US dollars [1]. Nearly 80% of persons with dementia living in the community require help with at least one activity of daily living (ADL) [2]. Persons with dementia requiring ADL assistance often exhibit resistance to care behaviors (RCBs) [3,4]. RCB refers to actions taken by a person with dementia to avoid receiving assistance or care activities [5,6,7]. RCB encompasses an entire spectrum of behaviors, from subtle verbal or nonverbal signals to full-on assaults [7,8]. Examples include pulling away or turning away from the caregiver, saying “no,” crying, yelling, pushing the caregiver away, grabbing an object to evade care, clenching the mouth shut (to avoid medication, food, or mouth care), adducting limbs (to prevent the axillae or perineal areas from being cleansed), and striking the caregiver [6,7,8]. These behaviors vary in frequency and intensity. Terms variously used to describe RCB include agitation, refusals, resistiveness to care, rejection of care, and care-resistant behavior [7,9,10]. For clarity, this article will use “resistance to care” behavior. Investigators in the past ten years have made important distinctions between agitation and resistance to care. The major difference between the two involves contextual clues. That is, agitation usually occurs without a specific trigger, whereas RCB occurs in response to a precipitating event, such as bathing or mouth care [7,11,12]. Agitation is evident in persons with mild dementia, increases with moderate dementia, and wanes as the dementia progresses to severe. RCB, however, consistently increases with worsening dementia severity; one study found an eightfold increase in overall RCB when dementia progressed to the severe stage [12].

Fauth et al. discovered that 66% of their 234 informal caregivers encountered RCB, the majority of which occurred when caregivers attempted to assist with ADL [3]. These informal caregivers experienced feelings of distress, captivity, and depression when faced with RCB. Shirai and Koerner [13] reported similar prevalence of RCB with fluctuations over time. They collected RCB data over an 8-day period from 63 informal caregivers and concluded that informal caregivers who experienced both higher frequencies and higher fluctuations in RCB experienced significant increases in physical problems such as headaches and chest tightness [13]. Spigelmyer et al. [14] conducted a phenomenological study examining the experience of informal caregivers with RCB; the investigators reported that caregivers felt intensely guilty and incompetent when faced with RCB [14].

In long-term care settings, Mahoney et al. were among the first to distinguish RCB from other behavioral and psychological symptoms of dementia (BPSD). Jablonski and colleagues have been instrumental in testing theoretically driven strategies that enable long-term care staff to successfully provide mouth care to persons with dementia exhibiting RCB [8,9,15,16,17]. For this study Jablonski et al.’s non-drug, behavioral interventions were adapted to a distance-accessible education, training, and coaching program for family caregivers of people with Alzheimer’s disease. The coaching program contained both asynchronous content and six 60-min weekly real-time coaching sessions with family caregivers delivered via an internet-based platform. We distinguished teaching, which is content-focused and generalized, from coaching, which is process focused and specific to the individual caregiver/learner. This content represents one part of a larger research program measuring the impact of the coaching program on caregiver burden and quality of life for families caring for persons with dementia. By its nature, most qualitative research has reflective qualities, including this one. However, the primary aim of this paper is to describe the methods, including process and content, of the six weekly 60-min dementia coaching sessions that constitutes the primary intervention in the broader research program. This is done by using the interactions between coach and caregiver (using actual quotes) as exemplars of the process and the content of this process.

## 2. Operationalization of Coaching Sessions: Planned Process and Content 

After receiving ethical approval from both the university (University of Alabama at Birmingham, UAB IRB-160819003) and the funder (United States Department of Defense, DOD HRPO A-19729) family members of persons with dementia were recruited from outpatient clinics within the same healthcare system. The family member participants were randomized to either immediate or delayed coaching sessions. All had immediate access to the asynchronous materials available on a password-protected website. The asynchronous content included six brief videos that provided and illustrated specific strategies for preventing and managing RCBs. The strategies have been published in-depth elsewhere [9,15,16,17].

Two members of the team served as coaches (RJ and VW). One coach was a nurse practitioner and researcher who had several years of clinical experience extending Jablonski and colleagues’ RCB mouth care work to other ADLs, such as bathing and medication administration. The other coach was a medical sociologist with experience implementing RCB research. The protocol allowed either coach to conduct coaching sessions.

The caregiver coaching sessions were operationalized a priori to follow a systematic process. Each coaching session was digitally recorded. During the sessions, the caregiver was queried about any problematic CRB and the strategies (if any) deployed to address it. Additional details, such as timing context, and success (or failure) were sought. The coaches adapted specific strategies to personalize the techniques to the needs of both the caregiver and the care recipient. The coaches assessed the ongoing efficacy of previously used interventions, and the resulting behavior, during each subsequent coaching session. The coaching goal for every session was to help the family caregiver become more independent with using and modifying the RCB techniques. Throughout the 6 coaching sessions, the efforts of family caregivers were acknowledged, praised, and reinforced.

The internet-based Go-To-Meeting™ platform was selected so that participants did not have to purchase any software; they downloaded the free version and responded to a link sent by the research team. Prior to the first scheduled coaching session, VW conducted a 20-minute practice session with the participants, using the link to assist with any technical problems. Every participant had a unique Go-to-Meeting™ link.

To date, 26 family caregivers have completed 6 coaching sessions. Each coaching session was approximately one hour in duration. Because of the length and depth of each one-hour coaching session, five participants that represented the diversity of the participant sample in terms of sequential entry into the study, gender, ethnicity, and family role of the caregiver (e.g., spouse versus adult child) were selected for the purpose of describing the content and process of the coaching sessions. Schatzman and Strauss [18] note that the use of purposeful sampling is feasible when choosing participants because of practical options such as constraints of time, the research framework, and burgeoning interests/concepts during the research process. These participants were also selected based on the richness of the data collected in their sessions and also represented the trajectory and content of the majority of the sessions The notes taken by the coaches during all coaching sessions were also analyzed. A description of the general pattern of all six coaching sessions emerged from the notes, which are described in the Results section, below. This pattern then informed our decision to select, and analyze, transcripts from the second and fifth sessions from the 5 purposively selected participants. The rationale for selecting these sessions was that sessions 2 and 5 best reflected caregiver struggles, problems, strategies, breakthroughs, and insights. During Session 2, participants provided richer descriptions of the RCBs as compared to the initial session. Participants also shared helpful information about underlying sources of strengths and challenges, for example, how existing family dynamics could be both supportive and frustrating. The Session 5 reflected the culmination of the previous 4 coaching sessions and consistently resulted in “a- ha” moments of exceptional clarity and understanding of the content. By Session 5, participants had tried the strategies and were becoming more confident with their abilities to reduce and manage RCB. The participants were selected to represent

The transcripts from Sessions 2 and 5 of the five purposively selected participants were transcribed, 10 transcripts total. Content analysis was employed on the 10 transcripts to better describe the behavioral strategies developed between the participants and the coaches. Content analysis is a research method that is useful in conceptualizing the meaning and relationship of language in a text in order to derive meaning [19]. All 10 transcripts were coded for concepts that described the coaching process. This provided a systematic method for thematic development based on the presence of these concepts. All 10 transcripts were compared against the original digital recordings for accuracy. Corrections were made if mistakes were noted.

## 3. Results

### 3.1. Sample and Demographics

The sample (*N* = 5) included 3 women and 2 men (see Table 1). The average age of the participants was 65.4 years. All of the caregivers were in their seventies except for one who was 29. Two of the caregivers were adult children of the care recipient.

### 3.2. Roles of Primary and Secondary Coaches 

The analyses of the transcripts revealed that the coaching process evolved and individual roles of each coach emerged. Both coaches participated together in the coaching sessions. Initially, the plan was for VW to gain familiarity and experience by “listening in” to all 6 of RJ’s coaching sessions with the first participant, and then for both coaches to conduct sessions independently. For the sessions reported here, RJ consistently served as the primary coach and VW as secondary coach. Participants were also aware that both coaches were present during each session.

After each session with any participant, both coaches engaged in debriefing activities and brainstormed ideas for the next session, which were incorporated into joint notes. This practice was found to be very helpful for both coaches; they continued to work together with all subsequent participants. The secondary coach would “listen in,” and assist the primary coach with post-coaching debriefing. Feedback from the secondary coach served as an assessment tool for the session. The secondary coach was able to provide feedback for factors such as participant reaction, the trajectory of the sessions and provide affirmation for successful interactions. Debriefing was especially helpful to the primary coach since a majority of the interactions occurred between the participant and primary coach. The post-session debriefing also alerted both coaches to the maturation and evolution of the intervention that produced a growing collection of strategies.

During the shared sessions, the secondary coach (VW) would interject ideas if appropriate. Notes taken during the sessions were shared between coaches and informed the context of the next session. The notes served to remind coaches of previous sessions, provided insight into the sessions, and served as a “placeholder” for the next session. These notes were especially helpful when multiple coaching sessions with other participants were being conducted in the interval between sessions for any one participant. In the notes, coaches recorded problems described by the caregivers that were not easily addressed in the session and required further thought. Coaches would discuss the problem and work together, brainstorming, to provide unique/individualized strategies for the caregiver. After the first several sessions, it became apparent that directed homework was important for the participants. It served to encourage participants to implement the strategies thus providing the coaches an opportunity for feedback to the caregiver and to assess the efficacy of the strategies for the care recipient.

### 3.3. General Pattern of Six Week Caregiver Coaching Sessions

The following is a description of the pattern of content and coaching activities that emerged from coaching notes across all sessions:

Session 1 served as the introduction and goal-setting session. The primary coach first inquired about the premorbid personality, work history, and interpersonal attributes of the person with dementia. The coach also inquired about past and current living situations and premorbid relationship quality. This information helped the coaches to assess the strengths and challenges presented by the relationship between the caregiver and care-recipient. Goals were negotiated and determined by asking the caregiver, “Which refusal behavior is most important for us to help you with?” The coaches laid the foundation for the future sessions by emphasizing that the coaching sessions were to help caregivers “manage” and “reduce” negative impacts of RCB on their well-being; because of the progressive nature of dementia, eradication of BPSD was not a realistic goal, and any expectation of eradication would set the caregivers up for failure. Session 1 also included the most teaching of dementia content.

Initial RCB strategies were also introduced during the first session and tied to the “shrinking box” and “short-staffed” analogies; a description of these and other analogies can be found in Table 2. These strategies included communicating using “short, sweet, and concrete” sentences; using gestures and pantomimes as adjuncts to verbal communication; avoiding arguments; and “entering their reality” in lieu of therapeutic fibbing. Therapeutic fibbing was a term coined in 1999 by Beach and Kramer, and referred to the practice of lying to people with dementia [20]. “Entering their reality” [15,17] involved finding situations from their past that would provide a reason for engaging in a care activity or allowing the caregiver to provide assistance without triggering refusals.

In Sessions 2–5, the coaches built on the initial RCB strategies *provided* during Session 1 while introducing new strategies as needed. Some content-focused and generalized teaching occurred throughout these sessions, but the amount of teaching progressively decreased while the amount of coaching increased (see Section 3.4 for specific coaching approaches). The caregiver was initially queried about the use and outcome of specific RCB strategies. The strategies were then tailored to fit unique situation of the caregiver and care-recipient. The coaches provided tailored “scripts” for the informal caregiver via role-playing, in which the coach assumed the role of the informal caregiver while the caregiver assumed the role of the person living with dementia. Throughout these five sessions, the coaches also validated and affirmed the informal caregiver’s techniques and encouraged continued adaption and tailoring of general RCB strategies. Every session concluded with a “homework” assignment based on the management of the behaviors thus far. The subsequent session began with a report on the efficacy of the homework assignment.

Although the focus was on the RCBs, it became obvious during the first few sessions that participants often asked questions about medications and other behaviors that occurred in addition to the RCB. Even though the focus was on RCB, the family caregivers were frequently and inadvertently triggering RCB. Thus, it became important for the coaches to discuss caregiver and care-recipient interactions in order to avoid triggering RCB.

The final session, Session 6, served as a review of successful strategies for managing RCB and also as the conclusion of the coaching relationship.

### 3.4. Four Themes of Coaching Content

In the 10 transcripts, four primary themes emerged from qualitative analysis of the coaching process: (1) education; (2) caregiver communication; (3) affirmation of the caregiver; and (4) individualized strategies. These four content categories were used throughout the coaching process and were interwoven with each other so that the participant knew why the behavior was occurring, how to verbally address it, how to use a strategy effectively, and affirmation of the result. The next sections include descriptions of each component of the process, and it provides examples. Direct quotes (some lengthy) from transcripts are used to illustrate typical coaching approaches.

#### 3.4.1. Education

Even though we differentiated teaching from coaching, the coaches provided each participant with a foundation for understanding the disease process that was systematic and continuous throughout the coaching sessions. This served to provide the caregivers with a “why and what” of care resistance. They provided a basic and succinct explanation of the disease process beginning with the neurobiology of threat perception (for more information, see Jablonski et al. 2011). Information provided was tailored to the background of caregivers. For example, two of the caregivers had advanced degrees in the biological and social sciences respectively so greater detail was provided when discussing the neurobiology of threat perception. Analogies were especially useful in explaining specific concepts for all caregivers. These are listed in Table 2. Neurodegeneration, for example, was explained using the analogy of the shrinking box:
“*The brain is like a box full of Christmas decorations. The decorations from 1964 are at the very bottom, while the decorations from last year at sitting at the top. As the box shrinks, the decorations from last year fall out. Meanwhile, you can more easily reach into the box and pull out the decorations from 1964 because you have fewer layers to move aside. The brain is like the box. As brain cells die, newer memories “fall out” while older memories are more accessible. That is why your family member cannot remember if she ate breakfast, but can tell you about some event 40 years ago. This is also why long, drawn out explanations and sentences do not work. The brain does not have enough space to ‘hold onto’ the entire conversation*.”

Caregivers often reported that their family members had episodes of irritability and anger; they shared their frustration that the person with dementia did not respond to reason. An analogy of a business or factory with overworked employees was used to explain how a reduced number of neurons could cause frustration and agitation in someone with Alzheimer’s disease.
“*Imagine if you showed up to work and half of your coworkers had just quit. You and your remaining colleagues would have to scramble and do two- or three-times the amount of work you usually do. You would be expected to perform tasks for which you have had little, or even no training. You and your colleagues may be able to compensate for an hour or two. By lunchtime, all of you would be overwhelmed, tired, and cranky. Customers would notice mistakes; they would notice orders being incorrectly filled or not completed at all. If a customer tried to explain something to you or one of your overworked colleagues, you may not have the energy to follow the conversation and may become even more irritated. By the end of the shift, all of you would be exhausted. That is what is going on in your loved one’s brain. Brain cells are trying to compensate and work harder. These hard-working cells become overwhelmed, especially in loud or crowded social situations. That is why your family member becomes irritable and may demonstrate anger, especially when you are trying to reason with them*.”

Caregivers often conveyed that their family member would refuse care and make statements such as, “I’ve already bathed,” when this event had not occurred. The coaches built on the analogies of “shrinking box” and “overworked staff” to help caregivers understand the confabulation that was often associated with care refusal behaviors. The coaches explained that the care recipient’s sensation of time passage was impaired because of the combination of short-term memory loss and more readily accessed older memories, which might cause a long-ago memory of bathing to feel as if the experience had just occurred. The coaches couched this explanation within the context of comparing the brain to a compartmentalized box; as the “walls” of the box breaks down, the contents (memories) become mixed up and tangled. Furthermore, cognitive slowing was explained using a highway metaphor: as neurons die, the lanes of the highway decrease and the brain attempts to accesses alternative routes the way motorists may access detours. Just as detour routes can become crowded and the traffic moves slower, the brain’s use of alternative (and less developed) networks can delay recall and slow down the ability to perform a desired action. Sometimes, these analogies were insufficient. In those cases, a new analogy of a boat without an anchor was introduced to help describe why persons with dementia may assert that an event had recently occurred when it had not.

Simple language to explain difficult concepts was useful in helping the caregiver understand a behavior with which they were struggling. The coaches also used the analogy of “moving backwards in time” to explain the person with dementia’s problems with praxis. New analogies often incorporated previously used analogies:
Coach (to Caregiver A:) “As she’s moving backwards in time, she may have forgotten about credit cards. ‘Cause when you forget stuff, you forget stuff the reverse way you learned it. It’s like the box that’s shrinking. The newest stuff on top is the first to fall out. She probably started using credit cards in later adulthood.”

While educating the caregiver was a significant portion of session 1, education was interwoven into most ongoing interactions with the caregivers. A context for why the behavior might be occurring was combined with an immediate strategy, or what to do, to address it.
Coach to Caregiver B: “Ok, you’re getting angry because you have to repeat yourself constantly.”Caregiver B: “Yeah, that’s right.”Coach: “Right, she can’t help it...her brain is shrinking so she has no place to put the memory she’s making now.”

#### 3.4.2. Caregiver Communication

In session 1, informal assessments were made about the quality and type of communication between the caregiver and the person living with dementia. In this assessment, patterns of communication that could trigger RCB were noted. This included tone and pitch level of voice, arguing, or cajoling. Throughout sessions 2 through 5, the coaches presented caregivers with new patterns of communication that included using scripts to communicate and a way of redefining interactions. Scripts were based on the relationship between caregiver and care recipient. In one interaction with caregiver C, the Coach stated,
Coach: “It’s the truth, it’s genuine, it’s loving. Those are the three components of the communication that are so important. It takes practice to, what I call, (develop) the scripts.”

The coach then provided him with a script to encourage his wife to eat. The strategy was to get her to come to sit with him and ultimately, eat.
Caregiver C: “How do I say, ‘it’s time for lunch?’ Her response is always, “I’m not hungry.”*Coach* (providing script to Caregiver C): ‘*I know you’re not hungry, but I would love your company while I eat*.’

In another interaction in the same session with the same caregiver, the coach asserted the importance of redefining modes of interaction as a way of more effective and less provocative communication. The caregiver was struggling with the strategy of “entering their reality,” where caregivers are coached to provide a dementia-centric rationale for accomplishing a task that is being resisted by the person with dementia. During this session, Caregiver C’s wife (the person with dementia), joined him in front of the computer screen and began interacting with the coach about the numerous bathrooms in their house. This comment prompted Caregiver C to contradict his wife and the Coach to intervene:
*Wife: There’s so many bathrooms in that house*.*Caregiver C: Ain’t but two bathrooms in that house*.*Coach* (responding to Caregiver C): *Okay, [*Name*]. Right there. Stop. Don’t argue. Because of the changes in the brain, every time she saw the same bathroom it was like a new bathroom. In the future, if she says something like, “Oh, there was a whole bunch of bathrooms,” you don’t have to lie, or you can just say, “Boy, it sure felt like that.” You’re not arguing. You’re acknowledging what she said. It’s factual from her perspective. Just because a person sees something a certain way doesn’t necessarily mean it’s wrong...You can explain, but in your wife’s situation, she doesn’t have enough brain power to process the explanations, and every time your wife says something and you present reality, it feels to her like you’re picking on her, you’re arguing. That’s going to increase her agitation. It’ll start low in the morning and it’s gonna build up. By suppertime, as the evening wears on, she may be very agitated, hard to settle down, and just into everything*.*Caregiver C: I feel like I’m on an edge or on a line that I don’t want to cross and become combative or ugly or dictatorial. I want to be real and not manipulative although it is. I’m just really struggling*.*Coach: You’re presenting things in a way that makes sense [to the person with dementia]. That’s not manipulation. It’s as if I go into the clinic and I have patients who speak Spanish. Okay, I can speak Spanish. Not well, but I can. I start talking to them in Spanish. That’s not manipulation. That’s communication. It’s crossing the [communication] bridge. For me to expect that person to communicate in English when so much is going on is a little unfair. I communicate in Spanish, even if I screw it up*.

#### 3.4.3. Affirmation of the Caregiver

It was important for the caregivers to believe that they were doing a good job. Caregivers often reported guilt or a sense of ineptitude in addressing RCB, as well as the negative emotions such as anger and resentment directed toward the care recipient. Affirmation was important and was used often. Caregivers were assured that they were being successful even when they had failed to use a discussed approach or strategy. The coaches also helped caregivers with setting realistic caregiving goals within the affirmations. Acknowledgement of their feelings in tandem with encouragement was also important. The following quotes illustrate this.
Coach to Caregiver B: A lot of care partners have this nagging concern. What if I don’t know something?What if I can’t do something? What if I make it worse? Honestly, I understand the littlevoice in your head saying those things, but it’s BS. You have a lot of knowledge in you*that you can adapt for this new journey, and you are successfully doing it*.Caregiver D: “Some of it is with my mind frame too; knowing that I’ve got to change that too. It’sjust the way that it is going to be from here on out. That’s part of it; knowing that I amgoing to have to change how we do things and when we do things andall of that. It’s just accepting that and finding the best way to do it.”Coach to Caregiver D: “Okay. A lot of people never get to that point. That’s good that you are at thatplace where you are really taking stock of how this is all going to work and you’refiguring it out.”

#### 3.4.4. Individualized Strategies

The coaches incorporated previously published strategies developed for institutional paid caregivers to prevent and manage RCB within the context of mouthcare to the coaching of the current study’s participants for assisting with other ADLs. These strategies included priming (using the environment to “trigger” procedural memories); distraction; chaining (caregiver begins the activity and allows the care recipient to finish the activity); bridging (caregiver has the person with dementia hold an object related to the caregiving activity, such as a wash cloth or soap during bathing); gentle touch; caregiver exhibiting happy/smiling facial expressions; caregiver speaking in short, respectful, 1-step sentences; caregiver avoiding excessive explanations; caregivers avoiding elderspeak (baby talk); and caregivers allowing as much self-care as safely possible [15,16,17]. The coaches consistently described the strategies as grounded in neurobiology. Placing behaviors in a physiologic context removed the personalization felt by caregivers; that is, caregivers often shared their perceptions that the person with dementia was rationally and purposefully engaging in a negative behavior. The coaches explained, using analogies, that many of the RCBs resulted from neurodegenerative changes:
*Coach (to Caregiver B): Okay. The communication thing, does that make sense? Shorter sentences, at least try to look less scary, a little smile, and gestures and pantomime can also reinforce [the message]. If I were to say to your wife, “Take off your glasses,” I would say, “Take off your glasses” [Coach pantomimed removing glasses] like that. Or if I was gonna say, “Brush your hair,” I’d say “Brush your hair” Of course, I’d have a brush or I’d mimic a brush brushing my hair. The gestures and pantomime can reinforce what we’re saying because there’s pieces of the brain right here in the temporal lobes, and those pieces of the brain, they take sound and put meaning towards it. They take words and assign meaning. As those sides of the brain shrink, words pop in, and the brain looks at the word and says, “I don’t know what to do with this.” The word just gets dropped. By communicating in layers and putting the layers on top, you’re more likely to get the message across*.*Caregiver B: I agree. I have found that gestures work a lot better, particularly in the—when there’s a sequence of events that you’ve got to do*.

Strategies to address resistance to care behavior were personalized and adapted to the specific context and needs of both caregiver and care recipient. At each session, caregivers were asked if the suggested strategies had been effective in managing or reducing the RCB. The caregiver below had success in using distraction and bridging to keep her mother seated on the toilet:
Caregiver E: *“last week you told me to put something in her hand to distract her from being angry... or to distract her from standing up. She’s been sitting down to go to the bathroom.”*

While the strategy of having her hold an empty prescription bottle worked initially worked, it was further adapted to be more successful for the care recipient. At times, the care recipient would throw the bottle on the floor and try to get up. The Coach suggested that Caregiver E put something in the bottle such as M&Ms or Tic Tacs. The Coach then explained that the sound of the candy rattling in the container might pique E’s mother’s interest and allow her to complete the task.

Caregivers were also coached to provide the person with dementia purposeful activities, which could then be interwoven into methods for preventing and managing future RCBs. Caregiver A was struggling with her mother’s refusal to remove clothes and to bathe. Caregiver A shared that her mother had worked as a housekeeper throughout her life. The Coach suggested that Caregiver A “hire” her mother to do the laundry and start by taking off her clothes to make a full load. Caregiver A successfully used this strategy:
Caregiver A: *I gave her some clothes to fold up*.Coach: *All right. How’d that work*?A: *She said give them—it worked out fine*.Coach: *Oh, really*?A: *I had to pay her but she folded ‘em up real neat*.Coach: *She was pretty happy about that*?A: *Yes, she was happy with that*.

The Coach and Caregiver A also discussed how caregivers’ emotions can impact the behavior of persons with dementia. This strategy became known as “the vibes,” and was often used within the context of helping caregivers to assist the care recipient with finding appropriate and meaningful activities to minimize boredom, another RCB trigger:
*Caregiver A: She used to make these little dolls and things like poodles out of the yarn. I tried to get her to remember that and she can’t. I might try it again but she made beautiful things like that*.Coach: Well ma’am, the goal is not for her to remember how to do it ‘cause she may or may not. The goal is for her to sit and feel a sense of accomplishment. Because sometimes we get hung up on the end product because it’s normal… What you can do is get the dementia out of the way so her personality can get through the cracks. She still has the same need to be loved, to be respected, to be taken seriously, to be treated with respect and so those things don’t change. Us wanting our loved ones to go back to making the bread they made or the poodles or the doll baby clothes, we’re just setting ourselves up to be frustrated. If you’re frustrated, what happens to your mom?*A: She gonna read them vibes and she’ll get frustrated*.*Coach: Right and now you got a mess. I wanna keep you from—I want you to enjoy the ride, enjoy the journey and accept what your mom can do, what she can’t. I’m not saying give up. I’m just saying switch up some of the expectations*.

The coaches explained to caregivers that their emotions, such as anger or frustration, could be felt by the person with dementia and trigger RCB. This was also described as “the vibes.” Caregiver B realized that he was acting more like a parent than a spouse because of his caregiver role, causing his “vibe” to change:
*Coach: Well, the other thing is, and this goes back to your wife, a kind word, a gesture, a compliment, that goes a long way too. Sometimes when we’re caring for people with dementia we’re so wrapped up in, we’re constantly doing for them, that we forget that they would like a compliment or to be treated less like a care recipient and more like a partner*.Caregiver B: True. I’ve seen that.Coach: When that dynamic changes, sometimes the spouse that is now—the spouse may not be able to articulate it, the care recipient spouse, but that spouse will often start to act up and will start to accuse the caregiver spouse of affairs, of not loving them, and what has happened is the vibe has changed. Some people don’t care. They feel the shift, and they go, “it still feels good.” Other people feel the shift, and they go, “Something’s not right.”*B: That’s a good way to explain that, and I can absolutely say that I have seen that. I didn’t know what I was seeing. I wonder if that’s what last night was*.

## 4. Discussion

This paper illustrates both the process and content of an online dementia coaching program for family caregivers of persons with dementia. The coaches successfully translated strategies developed to allow paid institutional caregivers to prevent and manage RCB within the context of mouth care to family caregivers to facilitate other ADLs in the home environment including bathing, dressing/undressing, medication administration, and eating. The coaching content was observed to fit in 4 categories: (1) education; (2) caregiver communication; (3) caregiver affirmation; and (4) individualized strategies. Even though the categories describe discrete themes, the coaching interactions usually addressed them concurrently rather than sequentially. These four content categories were used throughout the coaching process and were interwoven with each other so that the participant knew why the behavior was occurring, how to verbally address it, how to use a strategy effectively, and affirmation of the result. We were unable to find similar descriptions of caregiver coaching in the literature. However, Boots et al. [21], conducted a systematic review of 12 studies that used internet-based interventions to improve outcomes for informal caregivers of persons living with dementia; they concluded that the most successful interventions contained tailored information and used coaches to guide the use of the information.

Many of our caregivers were taught “therapeutic fibbing” from support groups and other dementia professionals. Therapeutic fibbing is lying to a person with dementia in order to obtain cooperation. We preferred the gentler strategy of “entering the care-recipient’s reality.” Caregiver A entered her mother’s reality and “hired” her mother to assist with the laundry; Caregiver A’s mother was a retired housekeeper. Caregiver A first told her mother to remove soiled clothing in order to have sufficient garments to make a full load of laundry. She asked her mother to fold clothes taken from the dryer. Caregiver A paid her mother once the clothes were folded. By entering her mother’s reality, Caregiver A was able to accomplish the important task of removing her mother’s soiled clothing in a respectful and dignified way, and she also provided her mother with a meaningful activity. When caregivers used entering their reality instead of therapeutic fibbing, the caregivers affirmed the care recipients’ emotions instead of offering a glib response. Entering their reality was also another way to avoid provocative communication. Caregiver B, for example, was vexed by his wife’s assertion that their house had more than two bathrooms and he corrected her during a coaching session. The Coach used entering their reality to illustrate to Caregiver B how he could validate his wife’s perception with a statement such as, “it sure felt like that.” The Coach explained how entering their reality can be used to acknowledge the care recipient’s reality.

The coaching process and content as described here is aligned with person-centered practices. Person-centered care recognizes the individual person’s self-determination, choices, worth, histories and interests [22]. The focus of care shifts from de-contextualized outcomes to those that are important to individuals. For people living with dementia, these outcomes often concern quality of life or the ability to function or care for themselves. Nolan and colleagues [23] took this idea one step further and proposed a relationship-centered model of care that includes not only the care recipient but also all who are involved in the care relationship. Their model’s underlying assumption is that every individual in the caring relationship should experience a sense of worth, purpose, and achievement. This approach is particularly salient for informal caregivers of persons who exhibit RCB and present safety concerns for all involved. The coaching intervention described here clearly emerged to be aligned with Nolan and colleagues’ relationship-centered model of care. The coaches inquired about the care-recipient’s past history, behavioral patterns and preferences; they helped the informal caregivers facilitate personalized relationship-based care and prevent many RCBs [24,25].

Kales et al. suggested a framework for clinicians to determine the etiology and optimal management of neuropsychiatric behaviors exhibited by persons with dementia [26]. This framework employed the mnemonic DICE: a description of the behavior; investigation of possible underlying causes of the behavior; creation of a behavioral plan; and evaluation of the behavioral plan [26]. Even though there was no a priori decision to implement the DICE framework, the natural progression of the coaching sessions mirrored the 4 steps.

The coaching intervention exhibited a maturational process as the study progressed. The coaches built on previous metaphors and analogies, not only within the six coaching sessions, but also with each successive participant. Their repertoire of behavioral strategies also increased as the study progressed. Maturation of behavioral interventional strategies has been identified by one other group of researchers [15]. The evolution and maturation of behavioral interventions may affect internal validity as a study progresses; if not reported or described, such maturations may explain subsequent problems with reproducibility.

We limited our thematic analyses to 10 coaching sessions involving 5 family caregivers. Our understanding of the coaching content would be enhanced by analyzing additional coaching sessions with participants near the end of the study, in order to capture more strategies and educational analogies as they evolve. Another limitation of our design was restricting participation to one family caregiver. Nearly every participant was assisted by other informal family caregivers. We plan to evaluate the efficacy of an online coaching program delivered to multiple family members.

The informal feedback from our participants has been favorable. The information captured from this methods paper will inform the development of a coaching manual and training materials. Given the growing number of persons with dementia living in the community, there is a pressing need for ongoing coaching programs that can be delivered by community laypersons and delivered in cost-efficient modalities.

## 5. Conclusions

Caregiver education, caregiver communication, caregiver affirmation, and individualized strategies are the foundational components of a successful dementia coaching program. All four components occur concurrently during coaching sessions; the amounts of the components differ as coaching sessions progress. The participants understood the etiology of refusal behaviors and were equipped with strategies to effectively manage those behaviors.

## Figures and Tables

**Table 1 healthcare-07-00013-t001:** Demographics.

Participant/Caregiver ID	Age	Gender	Race	Relationship to Care Recipient
**A**	75	F	African-American	Daughter
**B**	74	M	White	Spouse
**C**	78	M	White	Spouse
**D**	71	F	White	Spouse
**E**	29	F	Asian	Daughter

**Table 2 healthcare-07-00013-t002:** List of Analogies and Description.

Analogy	Purpose	Description
Shrinking box	Explanation of neurodegeneration and its impact on short-term memory	Coaches compared shrinking brains to shrinking boxes. As the boxes became smaller, recent memories fell out while new memories could not be placed in the shrinking box.
Short-staffedworking conditions	Explanation for periods of irritability and fatigue observed in the person living with dementia	The ongoing loss of neurons from neurodegeneration resulted in compensation from remaining neurons. These remaining neurons were likened to employees working with insufficient staff: the remaining employees would be expected to perform multiple and unfamiliar tasks, resulting in mistakes and fatigue. They would be irritable and cranky by the end of the shift.
Messy closet/Jewelry Box	Explanation for confabulation or erroneous long-term memories; also used to help explain altered sensation of time passing	The brain was compared to a compartmentalized closet or box, where memories were logically arranged in sequential patterns. If the clothes became haphazardly arranged in the closet, or if the compartments in the jewelry box disappeared, the contents would become disorganized and difficult to locate. This analogy was then applied to the brain, where the loss of neurons contributes to the jumbling of memories. The person with dementia was not lying or trying to be difficult.
Highways and back roads	Explanation for cognitive slowing and altered sensation of time passing	Neural networks were compared to highways. Loss of neurons resulted in “closed lanes” and “detours.” Both created traffic slowing. Ongoing neuronal death resulted in some highways becoming completely severed. The memories may still be available, but unable to be accessed. In these situations, sensations (music, smells, touch, pictures) may access specific memories via “back roads.”
Boat without an anchor	Additive explanation for confabulation or erroneous long-term memories; also used to help explain altered sensation of time passing	Short-term memory is necessary for people to “be anchored” in time and place. If short-term memory is impaired, the person with dementia becomes disoriented because they are like a boat without an anchor; they are bobbing around in a sea of memories without landmarks.
Moving backwards in time	Explanation for apraxia around ADLs and other activities.	Persons with dementia generally experience apraxia in a sequential pattern, with loss of complex abilities occurring prior to loss of simpler abilities. The loss of abilities was couched as reverse chronological aging. People lose the ability to use objects or perform activities that were learned latest in life, while retaining abilities (such as feeding onself) learned early in life.
Brain as computer	Explanation to assist the caregiver to understand the person’s inability to create new memories.	Loss of neurons compared to an older computer with no more memory.
Strategy Toolbox	Explanation of why strategies may require modifications over time and	Strategies compared to different tools in a toolbox. Just as every tool has a specific function, specific strategies are used in specific situations.

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
