# Peer review of "Description of Process and Content of Online Dementia Coaching for Family Caregivers of Persons with Dementia"

_healthcare, 2019, doi:10.3390/healthcare7010013_

Reviewer 1 Report

The article addresses a distinctive topic area and provides some well-written insights into dementia and care. I feel that the paper would be enhanced by a clearer alignment between the research question, the methodology and the findings.

A key challenge when reading the paper is establishing the purpose of the sample of participants. The research aim is well-stated in the introduction, but as the paper progresses it appears that the focus of the paper will be the experience of the sample members (and their interactions with the coaches). Instead, the emphasis is placed quite substantially on the role of the coaches. For example, the first theme ‘Education’ (p.7) principally addresses the input of the coaches, while the third theme ‘Affirmation of the caregiver’ (p.9 to p.10) includes one quote only and this is from a coach. The coaches (i.e. the authors) are themselves key members of the sample. This underscores that this is a reflective paper rather than a qualitative study as such. The discussion of the methodology generates the impression that carer-driven qualitative data will be the starting point for the findings. The reader’s expectations could be more effectively managed and it would also provide a more appropriate orientation to the subsequent elaboration of the identified themes.

Chapter two could accordingly be adjusted so that the emphasis on reflection on practice is rendered clearer. This seems to be central to the paper, with the selection of the sample providing focus for the reflection on the coaching process (rather than providing the centrepiece of a qualitative study into their experiences).

The paper also becomes rather difficult to follow with 26 caregivers completing six sessions and then five participants being selected as the analytical focus from the second to fifth session. There is quite a lot for the reader to take in here and the methodological process needs to be as lucid as possible. As the paper moves into the findings, it needs to be clear what stage of the research is being described. For example, the paper moves on to describe the sample characteristics of the five purposively selected individuals, but the first stage of the findings addresses session one, which I think refers to the research stage where all 26 participants were involved. The reader requires some additional assistance here to follow the process. In addition, how are the two sets of findings (from session one, and session two to five) complementary? These stages could be linked more clearly in the findings and also addressed more directly in the discussion.

The selection of the sample of five participants could be more clearly justified to avoid any sense of cherry-picking. The sentence explaining the choice of this purposive sample (line 108 to 111) could be expressed more effectively.

‘All 10 transcripts were coded for predefined concepts’ (line 116) requires a degree of elaboration. It is not clear what this means.

One of the caregiver quotes is not aligned with a particular participant (p.8). There is also some inconsistency; for example, sometimes ‘Participant C’ is used, while on other occasions ‘Caregiver C’ is employed (p.9).

It is notable, considering the small sample, that participant D is not associated with any of the quotes.

The themes could be represented much more effectively. The quotes from coaches in the education section are reasonable but generic. Why do these need to be aligned with a purposive sample? They appear to be general points that could apply to any of the sessions and are potentially background information rather than a feature of an ‘emerging’ theme. Some elements of this theme are valuably highlighted however, such as the use of analogies, the use of simple language and embedding education into interactions. The identification of key elements could have been clearer across the four themes.

The quotes in the ‘caregiver communication’ section are interesting, but what are they informing the reader about this theme in particular? There isn’t much sense that this represents a bounded and coherent theme, but rather some quotes that sit within a very broad (and inevitable) feature of coaching sessions i.e. communication.

While each themes does not require equal attention, ‘affirmation of the caregiver’ is dealt with in a rather cursory manner. Further evidence is required to establish the significance of this theme. The quote does not appear to show an ‘interaction’ as stated (line 322).

Material on ‘therapeutic fibbing’ in the discussion is very interesting, but it isn’t clear how this is building upon the previous material. This appears to be new material that would be better suited to the findings.

A quote is provided from a person with dementia (line 295). Had they provided consent for their participation? The paper refers to recruitment of family members.

I believe this paper can make a valuable contribution to the literature corpus. I feel it could be enhanced however by addressing the following points.

- The emphasis on reflection on practice should be amplified
- The role of the sample in relation to this reflection should be clarified
- The insights from the stages of the research should be linked more clearly (i.e. session one, and then sessions two to five)
- The themes should be strengthened.
- Key dimensions of these themes should be more readily identifiable and will provide a stronger platform for the discussion.

Author Response

We want to thank the reviewers for the encouraging words about the manuscript and for their suggestions for improvement. We have made the revisions suggested by the reviewers or attempted to explain why certain elements were included. Revisions and explanations are bolded and indented. We believe that the reviewers have helped us to strengthen and clarify the manuscript. Thank you again.

Response to Reviewer Comments

Reviewer 1:

1: A key challenge when reading the paper is establishing the purpose of the sample of participants. The research aim is well-stated in the introduction, but as the paper progresses it appears that the focus of the paper will be the experience of the sample members (and their interactions with the coaches). Instead, the emphasis is placed quite substantially on the role of the coaches. For example, the first theme ‘Education’ (p.7) principally addresses the input of the coaches, while the third theme ‘Affirmation of the caregiver’ (p.9 to p.10) includes one quote only and this is from a coach. The coaches (i.e. the authors) are themselves key members of the sample. This underscores that this is a reflective paper rather than a qualitative study as such. The discussion of the methodology generates the impression that carer-driven qualitative data will be the starting point for the findings. The reader’s expectations could be more effectively managed and it would also provide a more appropriate orientation to the subsequent elaboration of the identified themes.

·         The purpose of the paper is to use quotes from rich coaching sessions that illustrate how we conducted the coaching and of what the coaching consisted. Because it is a qualitative piece, some of the results and the discussion had reflective components but the purpose of the paper is to describe rather than reflect on the coaching sessions. As you noted, “the emphasis is placed quite substantially on the role of the coaches.” This is correct; however, this is to provide specific, concrete information/examples of how we did the coaching rather than any emphasis on caregiver or coaching experience (a great idea for future publications). Hopefully this was clarified in lines 73-77.

2: Chapter two could accordingly be adjusted so that the emphasis on reflection on practice is rendered clearer. This seems to be central to the paper, with the selection of the sample providing focus for the reflection on the coaching process (rather than providing the centerpiece of a qualitative study into their experiences).

·         As noted above, our emphasis is not on reflection of practice but rather description of practice. Hopefully we have clarified this in the methodology section with the following:

·         By its nature, most qualitative research has reflective qualities, including this one. However, the primary aim of this paper is to describe the methods, including process and content, of the six weekly 60-minute dementia coaching sessions that constitutes the primary intervention in the broader research program. This is done by using the interactions between coach and caregiver ( using actual quotes) as exemplars of the process and the content of this process. 

 3: The paper also becomes rather difficult to follow with 26 caregivers completing six sessions and then five participants being selected as the analytical focus from the second to fifth session. There is quite a lot for the reader to take in here and the methodological process needs to be as lucid as possible. As the paper moves into the findings, it needs to be clear what stage of the research is being described. For example, the paper moves on to describe the sample characteristics of the five purposively selected individuals, but the first stage of the findings addresses session one, which I think refers to the research stage where all 26 participants were involved. The reader requires some additional assistance here to follow the process. In addition, how are the two sets of findings (from session one, and session two to five) complementary? These stages could be linked more clearly in the findings and also addressed more directly in the discussion.

·         The first section in the results is a general description of coaching patterns across 6 sessions. Content analysis was only used for transcripts 2 and 5, not 2-5 from which the four themes were derived. Hopefully, we have clarified this in the manuscript. See changes in lines 118-119, 134-141, and lines 188-189.

4: The selection of the sample of five participants could be more clearly justified to avoid any sense of cherry-picking. The sentence explaining the choice of this purposive sample (line 108 to 111) could be expressed more effectively.

We added the following to provide the rationale for the selection of the five participants.

·         Each coaching session was approximately one hour. Because of the length and depth of each one-hour coaching session, five participants that represented the gender, age, and racial diversity of the total number of participants were selected for the purpose of describing the content and process of the coaching sessions. Schatzman and Strauss note that the use of purposeful sampling is feasible when choosing participants because of practical options such as constraints of time, the research framework, and burgeoning interests/concepts during the research process. These participants were also selected based on the richness of the data collected in their sessions and also represented the trajectory and content of the majority of the sessions The notes taken by the coaches during all coaching sessions were also analyzed. A description of the general pattern of all six coaching sessions emerged from the notes , which are described in the Results section, below

5. All 10 transcripts were coded for predefined concepts’ (line 116) requires a degree of elaboration. It is not clear what this means.

The following change was made to improve  clarity.

·         All 10 transcripts were coded for concepts that described the coaching process (Lines 135-136)

6: One of the caregiver quotes is not aligned with a particular participant (p.8). There is also some inconsistency; for example, sometimes ‘Participant C’ is used, while on other occasions ‘Caregiver C’ is employed (p.9).

·         That quote is from a coach to Caregiver A. For consistency we changed all specific references from Participant to Caregiver.

7:  It is notable, considering the small sample, that participant D is not associated with any of the quotes.

·         We included a quote from Caregiver D (see lines 352-362).

8: The themes could be represented much more effectively. The quotes from coaches in the education section are reasonable but generic. Why do these need to be aligned with a purposive sample? They appear to be general points that could apply to any of the sessions and are potentially background information rather than a feature of an ‘emerging’ theme. Some elements of this theme are valuably highlighted however, such as the use of analogies, the use of simple language and embedding education into interactions. The identification of key elements could have been clearer across the four themes.

·         The quotes reflecting the education theme are meant to be both descriptive of information and analogies used and are individualized based on the context of specific caregiver issues. While we provided an overview of the disease process, education was interwoven into every session based on issues or questions from the caregiver. While there is a generic component to the analogies and information, the analogies in particular evolved from specific caregiver questions/issues that arose during the sessions.

9: The quotes in the ‘caregiver communication’ section are interesting, but what are they informing the reader about this theme in particular? There isn’t much sense that this represents a bounded and coherent theme, but rather some quotes that sit within a very broad (and inevitable) feature of coaching sessions i.e. communication.

·         All types of coaching involve discussions of communication. The communication between a caregiver and a person with dementia can trigger behaviors that range from mild care resistance to full blown rage based on factors such as premorbid personality and relationship history. Coaching caregivers involved highly specific/stylized types of communication including scripts, learning to use language/gestures/facial expression and learning to use affirming language to avoid triggering care resistance, agitation, or anger.

10: While each themes does not require equal attention, ‘affirmation of the caregiver’ is dealt with in a rather cursory manner. Further evidence is required to establish the significance of this theme. The quote does not appear to show an ‘interaction’ as stated (line 322).

·          We have added another quote that represents the type of affirmation we provided to caregivers. “Interaction” was changed to “quotes.”

11: Material on ‘therapeutic fibbing’ in the discussion is very interesting, but it isn’t clear how this is building upon the previous material. This appears to be new material that would be better suited to the findings.

·         This was included in the discussion because it illustrated how our coaching process and content was aligned with person-centered practices. It further described  how our content and process is tailored to the individual by recognizing their need for autonomy, recognition of their worth, and their histories and interests.

 12: A quote is provided from a person with dementia (line 295). Had they provided consent for their participation? The paper refers to recruitment of family members.

·         Yes. Consent was obtained from the legally authorized representative using a protocol approved by two institutional review boards.

Reviewer 2 Report

Well done qualitative study. It is a very interesting topic that brings awareness and knowledge to the care giving of people with Alzheimer's Disease at a time when the older adults or baby boomers will increase the rate as the population ages.  Many family members (usually first relatives) are not equipped to handle the challenges of caring for them.

Title:  Precise and clearly indicates the essence of the study. 

Abstract: Concisely explains the purpose of the study and provides an accurate description of the how the study was carried out including a brief description of the results. The main findings are the 4 themes identified (education, communication, affirmation, individualization).

Introduction: The introduction gave the reader a good description of the study and the background.  However, the reader found it necessary to read the introduction more than once, slowly, to get a good sense of the study.  There are abbreviations used that caused the reader to go back for clarification.

Literature review:  well done. 

Methods: The reader is interested in knowing the rationale for selecting only 5 participants out of 26 that participated in the study; just as there is a rationale provided for the analysis of the coaching sessions 2 and 5, a brief statement indicating the rationale for the selected participants can help the reader have a better understanding of this method.  Additionally, it may help the reader understand the demographics of the sample in the context of care giving and the role of culture.  For AA's and Asians for example, there is cultural expectation and the care giving demands are often without any formal supports for the care giver.  There are some minor spelling errors (lines 190 and 193).  Line 195 refers to "participants often asked questions".  To maintain  consistency, the reader suggests to replace participant with informal caregiver as it has been called throughout the manuscript.

Data Analysis:  well done.  it is rich and very descriptive of the 4 themes identified. 

Discussion:  well done.  The limitations of the study are briefly discussed and the implications for further research provided as well. 

Conclusion: well done.  

Author Response

We want to thank the reviewers for the encouraging words about the manuscript and for their suggestions for improvement. We have made the revisions suggested by the reviewers or attempted to explain why certain elements were included. Revisions and explanations are bolded and indented. We believe that the reviewers have helped us to strengthen and clarify the manuscript. Thank you again.

1.       The reader is interested in knowing the rationale for selecting only 5 participants out of 26 that participated in the study; just as there is a rationale provided for the analysis of the coaching sessions 2 and 5, a brief statement indicating the rationale for the selected participants can help the reader have a better understanding of this method. 

We added the following section to provide our rationale for the selection of only 5 participants

·         Each coaching session was approximately one hour. Because of the length and depth of each one-hour coaching session, five participants that represented the gender, age, and racial diversity of the total number of participants were selected for the purpose of describing the content and process of the coaching sessions. Schatzman and Strauss note that the use of purposeful sampling is feasible when choosing participants because of practical options such as constraints of time, the research framework, and burgeoning interests/concepts during the research process. These participants were also selected based on the richness of the data collected in their sessions and also represented the trajectory and content of the majority of the sessions The notes taken by the coaches during all coaching sessions were also analyzed. A description of the general pattern of all six coaching sessions emerged from the notes , which are described in the Results section, below.

Reviewer 3 Report

The reported results are informative and original. Very specific guidelines for coaching and helpful. Results interpreted appropriately, informative and conclusions justified. Authors discuss that small number of interviews. Will be interesting to see if analysis of further transcripts supports current conceptualizations. The authors state this.

Design and analysis appropriate

A couple of notes: 

Line 33 Literature refers to resistiveness to care, not rejection of care.

Others refer to rejection of care. Measure cited is resistiveness to are. Need to use consistent terms. 

Are authors using these terms interchangeably? If so need to state

Line 268 where is the “what to do to address it as stated above.

Table 2 especially helpful, as are several of the quotes from the caregiver and response from. coach

Author Response

We want to thank the reviewers for the encouraging words about the manuscript and for their suggestions for improvement. We have made the revisions suggested by the reviewers or attempted to explain why certain elements were included. Revisions and explanations are bolded and indented. We believe that the reviewers have helped us to strengthen and clarify the manuscript. Thank you again.

1.       Line 33 Literature refers to resistiveness to care, not rejection of care. Others refer to rejection of care. Measure cited is resistiveness to care. Need to use consistent terms. 

·         Thank you for your review of our manuscript and for your kind words. In the literature, these terms are often used interchangeably and represent degrees of resistiveness- from mild resistance to total rejection of care. We are using resistance to care to describe the concept.

2.       Line 268 where is the “what to do to address it “as stated above?

·         For clarity we have removed this statement from the manuscript.

Round  2

Reviewer 1 Report

The revisions have enhanced the paper and provided the necessary clarifications.